# Culture-independent detection and characterisation of *Mycobacterium tuberculosis* and *M. africanum* in sputum samples using shotgun metagenomics on a benchtop sequencer

Emma L. Doughty[1], Martin J. Sergeant[1], Ifedayo Adetifa[2], Martin Antonio[1,2] and Mark J. Pallen[1]

[1] Microbiology and Infection Unit, Warwick Medical School, University of Warwick, Coventry, United Kingdom
[2] Medical Research Council Unit, Fajara, The Gambia

Corresponding author
Mark J. Pallen,
m.pallen@warwick.ac.uk

## ABSTRACT

Tuberculosis remains a major global health problem. Laboratory diagnostic methods that allow effective, early detection of cases are central to management of tuberculosis in the individual patient and in the community. Since the 1880s, laboratory diagnosis of tuberculosis has relied primarily on microscopy and culture. However, microscopy fails to provide species- or lineage-level identification and culture-based workflows for diagnosis of tuberculosis remain complex, expensive, slow, technically demanding and poorly able to handle mixed infections. We therefore explored the potential of shotgun metagenomics, sequencing of DNA from samples without culture or target-specific amplification or capture, to detect and characterise strains from the *Mycobacterium tuberculosis* complex in smear-positive sputum samples obtained from The Gambia in West Africa. Eight smear- and culture-positive sputum samples were investigated using a differential-lysis protocol followed by a kit-based DNA extraction method, with sequencing performed on a benchtop sequencing instrument, the Illumina MiSeq. The number of sequence reads in each sputum-derived metagenome ranged from 989,442 to 2,818,238. The proportion of reads in each metagenome mapping against the human genome ranged from 20% to 99%. We were able to detect sequences from the *M. tuberculosis* complex in all eight samples, with coverage of the H37Rv reference genome ranging from 0.002X to 0.7X. By analysing the distribution of large sequence polymorphisms (deletions and the locations of the insertion element IS*6110*) and single nucleotide polymorphisms (SNPs), we were able to assign seven of eight metagenome-derived genomes to a species and lineage within the *M. tuberculosis* complex. Two metagenome-derived mycobacterial genomes were assigned to *M. africanum*, a species largely confined to West Africa; the others that could be assigned belonged to lineages T, H or LAM within the clade of "modern" *M. tuberculosis* strains. We have provided proof of principle that shotgun metagenomics can be used to detect and characterise *M. tuberculosis* sequences from sputum samples without culture or target-specific amplification or capture, using an accessible benchtop-sequencing platform, the Illumina MiSeq, and relatively simple DNA extraction, sequencing and bioinformatics protocols. In our hands, sputum

metagenomics does not yet deliver sufficient depth of coverage to allow sequence-based sensitivity testing; it remains to be determined whether improvements in DNA extraction protocols alone can deliver this or whether culture, capture or amplification steps will be required. Nonetheless, we can foresee a tipping point when a unified automated metagenomics-based workflow might start to compete with the plethora of methods currently in use in the diagnostic microbiology laboratory.

# INTRODUCTION

Tuberculosis (TB) is an infection, primarily of the lungs, caused by *Mycobacterium tuberculosis* and related species within the *M. tuberculosis* complex. TB remains a major global health problem, second only to HIV/AIDS in terms of global deaths from a single infectious agent—according to estimates from the World Health Organisation (WHO), 8.6 million people developed TB in 2012 and 1.3 million died from the disease, including 320,000 deaths among HIV-positive individuals (*WHO, 2013*).

Central to management of TB in the individual patient and in the community are laboratory diagnostic methods that allow effective, early detection of cases. Since the pioneering work of Koch and Ehrlich in the 1880s, laboratory diagnosis of pulmonary TB has largely relied on acid-fast staining of sputum samples and culture on selective laboratory media for the isolation of mycobacteria (*Ehrlich, 1882*; *Koch, 1882*). Microscopy is still generally used as a first-line diagnostic approach and as the only laboratory approach in resource-poor settings (*Drobniewski et al., 2012*) Smear-positivity is also used as a guide to infectivity and responsiveness to treatment. However, microscopy fails to provide species-level identification of acid-fast bacilli (*Maiga et al., 2012*). Such identification is important in guiding treatment, because pathogenic mycobacteria from outside the *M. tuberculosis* complex often fail to respond to conventional anti-TB treatment (*Maiga et al., 2012*). Furthermore, there are important differences in response to treatment even within the *M. tuberculosis* complex. *M. bovis* and *M. canettii* fail to respond to the first-line anti-tuberculous agent pyrazinamide—as a result, failure to recognise *M. bovis* as a cause of TB can have fatal consequences (*Allix-Beguec et al., 2010*). In addition, *M. canettii* appears to show decreased susceptibility to a promising new anti-TB drug candidate, PA-824 (*Feuerriegel et al., 2011*; *Feuerriegel et al., 2013*).

There is also increasing recognition of lineage- or species-specific differences in pathogen biology within the *M. tuberculosis complex. M. africanum*, which is largely restricted to West Africa, where it causes up to half of human pulmonary TB, is associated with less transmissible and less severe infection than typical strains of the "modern" *M. tuberculosis* clade (*de Jong, Antonio & Gagneux, 2010*). Similarly, *M. canettii*, restricted to the horn of Africa, and *M. bovis*, both usually a spillover from animals, transmit relatively poorly from human to human (*Fabre et al., 2010*; *Gonzalo-Asensio et al., 2014*).

By contrast, the Beijing-W lineage of *M. tuberculosis sensu stricto*, which has spread around the world in recent decades, appears to cause more aggressive disease and is more likely to become drug-resistant (*Nicol & Wilkinson, 2008*; *Borgdorff & van Soolingen, 2013*).

Owing to the slow growth rate of the *M. tuberculosis* complex, traditional culture-based diagnosis of TB typically takes several weeks or even months. Similarly, conventional phenotypic mycobacterial sensitivity testing remains slow and may not be reliable for all classes of anti-tuberculous agent. In recent decades, automated detection of growth in liquid culture, through e.g. the mycobacteria growth indicator tube (MGIT), has led to improvements in the speed and ease of diagnosis, so that diagnosis by culture is now often possible within a fortnight (*Pfyffer et al., 1997*).

However, in comparison to most other laboratory procedures, culture-based diagnostic workflows for TB remain complex, expensive, slow, technically demanding and require expensive biocontainment facilities. Furthermore, as isolation of mycobacteria in pure culture and sensitivity testing remain onerous, in resource-poor settings these steps are omitted and, even in well-resourced laboratories, typically only one or a few single-colony subcultures are followed up from each sample. This leads to under-recognition of mixed infections, where more than one strain from the *M. tuberculosis* complex is present or where TB co-occurs with infection by other mycobacteria (*Shamputa et al., 2004*; *Warren et al., 2004*; *Cohen et al., 2011*; *Wang et al., 2011*). This can lead to difficulties in treatment when strains or species susceptible to conventional anti-tuberculous treatment co-exist with resistant strains or species within the same patient (*Hingley-Wilson et al., 2013*).

As an alternative to culture and phenotypic sensitivity testing, the WHO has recently recommended a new, rapid, automated, real-time amplification-based TB diagnostic test, the Xpert MTB/RIF assay (*WHO, 2011*). This system allows simultaneous detection of *M. tuberculosis* and rifampicin-resistance mutations in a closed system, suitable for use in a simple laboratory setting, while providing a result in less than two hours directly from sputum samples (*Helb et al., 2010*). However, this approach performs suboptimally on mixed infections, fails to provide the full range of clinically relevant information (e.g., speciation, susceptibility to other agents) and, in sampling only a small fraction of the genome, affords no insight into pathogen biology, evolution, and epidemiology (*Zetola et al., 2014*).

Epidemiological investigation of clinical isolates from the *M. tuberculosis* complex plays an important role in the management and control of TB. A range of molecular typing schemes have been developed, including IS*6110* fingerprinting, mycobacterial interspersed repetitive unit-variable number of tandem repeat (MIRU-VNTR) and spoligotyping (*Jagielski et al., 2014*). These approaches can be valuable in distinguishing relapse from re-infection and in recognising mixed infections within the individual patient, as well as identifying sources of infection, detecting outbreaks and tracking spread of lineages within a community. However, as these approaches usually require isolation of the pathogen in pure culture, clinically relevant typing data is typically not available until 1–2 months after collection of a sputum sample.

Over the past fifteen years, whole-genome sequencing has been applied to a steadily wider range of isolates from *M. tuberculosis* and related species (*Cole et al., 1998*;

*Brosch et al., 2002*; *Gutierrez, Supply & Brosch, 2009*). These efforts have shed light on the evolution and population structure of this group of pathogens, showing that members of the *M. tuberculosis* complex are reproductively isolated, engaging in almost no horizontal gene transfer and showing a clonal population structure in which lineages diverge through a limited set of genetic changes, including point mutations, deletions, movement of insertion elements and rearrangements within repetitive regions. Whole-genome analyses allow isolates to be assigned to a range of species, global lineages and sub-lineages on the basis of single nucleotide polymorphisms (SNPs) and large sequence polymorphisms (typically deletions, which are often termed "regions of difference" or RDs, and insertion of the transposable element IS*6110*).

In recent years, the availability of rapid, cheap high-throughput sequencing and, particularly, the arrival of user-friendly benchtop sequencing platforms, such as the Illumina MiSeq (*Loman et al., 2012a*; *Loman et al., 2012b*), have led to the widespread use of whole-genome sequencing in TB sensitivity testing and epidemiology, with adoption of whole-genome sequencing for routine use in some TB reference laboratories (*Gardy et al., 2011*; *Koser et al., 2012*; *Roetzer et al., 2013*; *Walker et al., 2013*; *Walker et al., 2014*). However, high-throughput sequencing has not yet been used as a diagnostic tool for TB, because it has been assumed that one needs to subject clinical samples to prolonged culture before sufficient mycobacterial DNA can be obtained for whole-genome sequencing and analysis. Some researchers (*Koser et al., 2013*) have recently challenged this assumption by obtaining mycobacterial genome sequences from DNA extracted directly from a three-day MGIT culture of a sputum sample. However, this begs the questions: why bother with culture; why not obtain mycobacterial genome sequences directly from a sputum sample, without culture?

Shotgun metagenomics—that is the unbiased sequencing *en masse* of DNA extracted from a sample without target-specific amplification or capture—has provided a powerful assumption-free approach to the recovery of bacterial pathogen genomes from contemporary and historical material (*Pallen, 2014*). This approach allowed an outbreak strain genome to be reconstructed from stool samples from the 2011 *Escherichia coli* O104:H4 outbreak and has proven successful in obtaining genome-wide sequence data for *Borrelia burgdorferi*, *M. leprae*, *M. tuberculosis* and *Brucella melitensis* from long-dead human remains (*Keller et al., 2012*; *Chan et al., 2013*; *Loman et al., 2013*; *Schuenemann et al., 2013*; *Kay et al., 2014*). Metagenomics has recently provided clinically useful information in cases of chlamydial pneumonia and neuroleptospirosis (*Fischer et al., 2014*; *Wilson et al., 2014*).

Here, we explore the potential of metagenomics in detecting and characterising *Mycobacterium tuberculosis* and *M. africanum* strains in smear-positive sputum samples from patients from The Gambia in West Africa.

## MATERIALS AND METHODS

### Microbiological analysis and sample selection

Eight smear- and culture-positive sputum samples were selected for metagenomic analysis from specimens collected in May 2014 under the auspices of the Enhanced

Case Finding project (http://clinicaltrials.gov/show/NCT01660646). The joint Gambia Government/MRC Ethics Committee approved this investigation under reference SCC 1232 and informed written consent was obtained for all participants. The sputum samples were collected by expectoration into a sterile cup and transported on ice to the TB laboratory at the MRC Gambia unit within 24 h of collection.

Prior to selection for metagenomic investigation, an aliquot of each sample was subjected to microbiological analysis. These specimens were decontaminated by the sodium hydroxide and *N*-acetyl-l-cysteine (NaOH/NALC) method, with final concentrations of 1% for NaOH, 1.45% sodium citrate and 0.25% for NALC. Sputum smears were prepared by centrifuging 3–10 mL decontaminated sputum and then resuspending pellets in 2 mL buffer. Smears were stained with auramine-O and then examined by fluorescence microscopy. Positive smears were confirmed by Ziehl-Neelsen staining. 20–100 fields were examined at 1000X magnification and smear-positive samples were scored quantitatively as 1+, 2+ or 3+ (*Kent & Kubica, 1985*).The presence of *M. tuberculosis complex* in samples was confirmed by culture in the BACTEC MGIT 960 Mycobacterial Detection System and on slopes of Löwenstein–Jensen medium. Cultured isolates were subjected to spoligotyping as previously described (*Kamerbeek et al., 1997*; *de Jong et al., 2009*).

## DNA extraction using differential lysis

DNA extraction was performed in the TB laboratory in the MRC Unit in The Gambia. Aliquots of unprocessed sputum were subjected to a differential lysis protocol, modified from a published method for metagenomic analysis of sputum from cystic fibrosis patients (*Lim et al., 2013*). In this method, human cells are subjected to osmotic lysis and then the liberated human DNA is removed by DNase treatment. To monitor contamination within the laboratory, we processed two negative-control samples containing only sterile water via the same method.

At the start of the differential lysis protocol, a 1 mL aliquot of whole sputum was mixed with 1 mL decongestant solution (0.25 g N-acetyl L-cysteine, 25 mL 2.9% sodium citrate, 25 mL water) until liquefied and incubated for 15 min at room temperature. 48 mL phosphate-buffered solution (pH 7) was added and mixed thoroughly, before centrifugation at $3,220 \times$ g for 20 min. The pellet was resuspended in 10 mL sterile deionised water and incubated at room temperature for 15 min, so that human cells undergo osmotic lysis, while mycobacterial cells remain intact. The centrifugation and resuspension-in-water steps were repeated before a final round of centrifugation. The pellet was then treated with the RNase-Free DNase Set (Qiagen), adding 25 μL DNase I (2.73 Kunitz units per μL), 100 μL RDD buffer and 875 μL sterile water. The sample was then incubated at room temperature for 2 h, with repeated inversion of the tubes. The sample underwent two rounds of centrifugation and resuspension of the pellet in 10 mL TE buffer (0.01 M Tris–HCl, 0.001 M EDTA, pH 8.0). Finally, before DNA extraction began, the sample was centrifuged and the pellet was resuspended in 500 μL TE buffer. On completion of the differential lysis protocol, samples underwent heat treatment at 75 °C for 10 min, followed by DNA extraction using a commercial kit, the NucleoSpin

Tissue-Kit (Macherey-Nagel, Duren, Germany), according to the manufacturer's protocol for hard-to-lyse bacteria.

## Library preparation and sequencing

DNA samples were sent to Warwick Medical School, Coventry, UK, where all further laboratory and bioinformatics analyses were performed. The concentration of DNA present in each extract was determined using the Qubit 2.0 fluorometer and Qubit® dsDNA Assay Kits according to the manufacturer's protocol (Invitrogen Ltd., Paisley, United Kingdom), using the HS (high-sensitivity) or BR (broad-range) kits, depending on the DNA concentration. There was no detectable DNA in the negative control samples with the HS kit, which is sensitive down to 10 pg/μL. DNA extracts were diluted to 0.2 ng/μL and were then converted into sequencing libraries, using the Illumina Nextera XT sample preparation kit according to the manufacturer's instructions (Illumina UK, Little Chesterford, United Kingdom). The libraries were sequenced on the Illumina MiSeq at the University of Warwick.

## Identification of human and mycobacterial sequences

Sequence reads were mapped against the genome of *Mycobacterium tuberculosis* H37Rv (GenBank accession numbers AL123456) and the human reference genome hg19 (GenBank Assembly ID: GCA_000001405.1), using Bowtie2 version 2.1.0 (*Langmead & Salzberg, 2012*), using relaxed and stringent protocols. The relaxed protocol exploited the option `--very-sensitive-local`. The stringent protocol allowed only limited mismatches (3 per 100 base pairs) and soft clipping of poor quality ends, by exploiting the options `--ignore-quals --mp 10,10 --score-min L,0,0.725 --local --ma 1`. A custom-built script was used to convert coverage data from the BAM files into a tab-delineated format that was then entered into Microsoft Excel, which was then used to generate coverage plots. Metagenomic sequence reads from this study (excluding those that mapped to the human genome) have been deposited in the European Nucleotide Archive under the following accession numbers: ERS542292, ERS542293, ERS542294, ERS542295, ERS542296, ERS542297, ERS542298, ERS542299.

## Species and lineage assignment using low-coverage SNPs

For the phylogenetic analysis using SNPs, we selected representative genomes from each of the species and major lineages within the *M. tuberculosis* complex that infect humans, drawing on lineage designations reported by PolyTB (*Coll et al., 2014*). Genome sequences were taken from entries in the short read archive ERP000276 and ERP000124 (http://www.ncbi.nlm.nih.gov/Traces/sra/). We then mapped these genomes against *M. tuberculosis* H37Rv with Bowtie2 under default settings and then called SNPs using VarScan2 (*Koboldt et al., 2012*). Any SNPs that fell within a set of previously published repetitive genes were excluded from further analysis (*Comas et al., 2010*). SNPs were used to construct a tree with RAxML version 7 (*Stamatakis, 2014*), using default parameters with the GTR-gamma model. Reads from the metagenome from each sample were mapped against the reference strain *M. tuberculosis* H37Rv using the default settings in Bowtie2 and the majority base

| Table 1 | Sample characteristics and sequencing results. | | | |
|---|---|---|---|---|
| **Sample** | **ZN grade** | **DNA concentration in extract (µg/mL)** | **Total no. reads** | **% reads aligning to human genome** |
| K1 | 3+ | 27.8 | 989,442 | 73.71 |
| K2 | 3+ | 2.28 | 2,170,640 | 78.46 |
| K3 | 2+ | 71 | 1,617,808 | 99.3 |
| K4 | 2+ | 250 | 1,204,408 | 97.22 |
| K5 | 2+ | 7.7 | 1,537,676 | 74.17 |
| K6 | 2+ | 48.8 | 2,411,708 | 97.47 |
| K7 | 1+ | 25 | 2,818,238 | 50.59 |
| K8 | 1+ | 0.63 | 1,851,892 | 20.29 |

called from each SNP position with no quality filtering. If no base was present at the position, a gap was used. The pplacer suite of programs (*Matsen, Kodner & Armbrust, 2010*) was then used to assign the sequence to a species and lineage on the mycobacterial tree.

## Lineage assignment using IS*6110*-insertion-site profiles

We mapped each metagenome against the sequence of IS*6110* (Genbank accession number: AJ242908) using Bowtie's `--local` option, which performs a softclipping of the mapped sequences. We then extracted IS6110-flanking sequences by retrieving all sequences >30 bp that had that had been softclipped from the ends of the element. These sequences were then mapped against the H37Rv genome using Bowtie2 and the coordinates of the IS*6110* insertion points determined.

## RESULTS

### Detection of the *M. tuberculosis* complex in sputum samples

We obtained metagenomic sequences from eight smear- and culture-positive sputum samples. The number of sequence reads in each sputum-derived metagenome ranged from 989,442 to 2,818,238 (Table 1). The proportion of reads from each sample mapping to the human reference genome hg19 varied from 20% to 99%.

Coverage from reads mapping to the genome of the *M. tuberculosis* reference strain H37Rv under relaxed settings ranged from 0.009X to 1.3X (Table 2). However, we suspected that many of the matches represented false-positives. To confirm our suspicion, we calculated the average read depth at the positions where reads matched.

If the matches occurred because of sequence identity with conserved genes from other species, one would expect there to be multiple reads matching each mapped position, whereas for a shotgun library where the coverage is less than 1X, one would expect the average read depth to be around 1. However, as we created our sequence libraries using a paired-end protocol, there will be variable overlap between reads originating from the same DNA fragment, so one would expect the average read depth for a genuine random shotgun under these conditions to sit between 1 and 2. However, when mapping was performed under relaxed conditions, the average read depth was >2 in six of the eight

**Table 2** Mapping to *M. tuberculosis* H37Rv reference genome.

| Sample | Under relaxed mapping conditions | | | Under stringent mapping conditions | | |
|---|---|---|---|---|---|---|
| | Bases aligning to H37Rv | Coverage of H37Rv | Average read depth | Bases aligning to H37Rv | Coverage of H37Rv | Average read depth |
| K1 | 410,228 | 0.093 | 2.2 | 141,906 | 0.032 | 1.3 |
| K2 | 5,685,901 | 1.289 | 2.3 | 3,057,187 | 0.693 | 1.9 |
| K3 | 99,643 | 0.023 | 1.3 | 54,413 | 0.012 | 1.2 |
| K4 | 40,019 | 0.009 | 1.9 | 10,840 | 0.002 | 1.3 |
| K5 | 732,623 | 0.166 | 2.5 | 238,451 | 0.054 | 1.3 |
| K6 | 94,023 | 0.021 | 2.3 | 34,704 | 0.008 | 1.7 |
| K7 | 1,366,309 | 0.310 | 11.4 | 50,873 | 0.012 | 1.5 |
| K8 | 1,725,816 | 0.391 | 7.7 | 109,514 | 0.025 | 1.3 |

samples and in two cases was >7 (Table 2), indicating a major contribution from spurious matches to conserved genes.

To restrict matches to the H37Rv genome to genuine on-target alignments, we then mapped each metagenome against the reference strain under high-stringency conditions ($\leq$3 mismatches per 100 base pairs, with soft clipping of poor quality ends). This led to a decrease in reads mapping to H37Rv in all samples, with coverage of the H37Rv under stringent settings ranging from 0.002X to 0.7X. Nonetheless, we recovered between ~11,000 and 3 million base pairs of *M. tuberculosis* sequence from our samples under such stringent conditions (Table 2). The average read depth in the samples fell to between 1.2 and 1.9, consistent with expectations for a random shotgun (Table 2).

## Phylogenetic placement of *M. tuberculosis* strains using SNPs

Conventional phylogenetic methods based on identification of trusted SNPs cannot be applied to the kinds of low-coverage genome sequences we have obtained here. However, the technique of "phylogenetic placement" provides an alternative solution (*Matsen, Kodner & Armbrust, 2010*; *Kay et al., 2014*). Here, one draws on a fixed reference tree, computed from high-coverage genomes, and places the unknown query sequence on to the tree using programs such as pplacer (*Matsen, Kodner & Armbrust, 2010*). To perform phylogenetic placements on our samples, we derived a set of phylogenetically informative SNPs from representatives of the major lineages within the *M. tuberculosis* complex. We then analysed reads from each of the sputum metagenomes that aligned to equivalent positions in the H37Rv genome.

Using this approach, despite the low coverage, we could confidently assign (with a posterior probability of >0.97), all but one of the metagenome-derived mycobacterial genomes to a species and lineage within the *M. tuberculosis* complex (Fig. 1). In all these cases, the conclusions from metagenomics matched those from spoligotyping of cultured isolates (Table 3). For two of the samples (K3, K5), the metagenome-derived genome was assigned to *M. africanum* clade 2, which is consistent with the known high-prevalence of this lineage in The Gambia (*de Jong et al., 2010*). Five samples were assigned to the

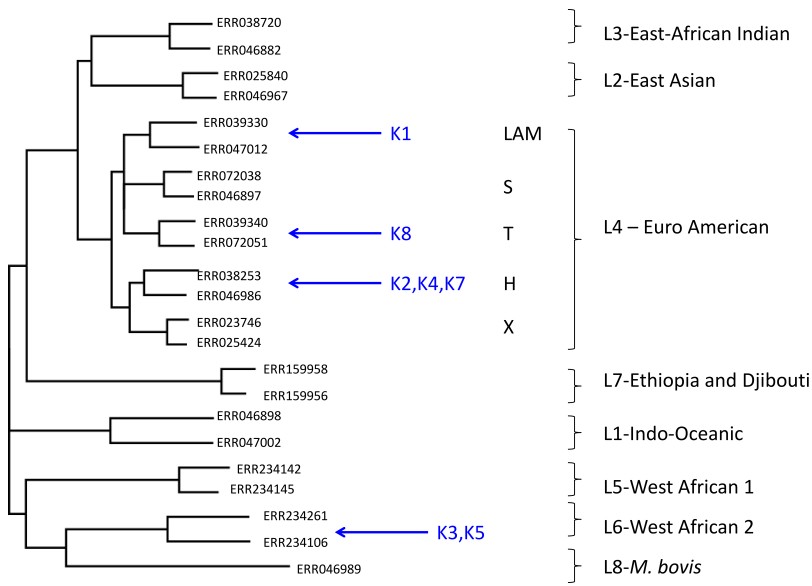

**Figure 1 Maximum likelihood tree showing placement of mycobacterial metagenome-derived genomes amongst the major lineages and clades within the *M. tuberculosis* complex.** Detection and characterisation of *Mycobacterium tuberculosis* in sputum samples using shotgun metagenomics. Two representatives from each lineage/clades are shown. Tree calculated using RaXML and rooted with *M. canetti* (not shown).

Euro-American lineage (also termed Lineage 4), which sits within the clade of modern *M. tuberculosis* strains and which is known to be highly prevalent in The Gambia (*de Jong et al., 2010*). Phylogenetic placement allowed three of these samples to be assigned to sub-lineage H, one to the T-clade and one to the LAM clade.

### Species and lineage assignment using IS*6110* insertion sites

From four samples, we were able to retrieve information on IS*6110* insertion sites (Table 4). In two of the three samples (K2, K4) assigned to the H clade by phylogenetic placement, we discovered IS*6110* insertion sites that had previously been reported as specific to the Haarlem or H clade (HSI1, HSI2, HSI3), thereby confirming the SNP-based lineage assignment (*Cubillos-Ruiz et al., 2010*). In the sample assigned to the LAM clade, we retrieved information on a single *IS6110* insertion site, which disrupts the coding sequence Rv3113. This insertion has been reported as specific to the LAM clade (*Lanzas et al., 2013*), again confirming the SNP-based lineage assignment. In one of the two samples assigned to *M. africanum*, we retrieved information on a single IS6110 insertion site. However, this insertion appeared to be absent from all other available genome-sequenced strains from the *M. tuberculosis* complex, so was phylogenetically uninformative.

### DISCUSSION

Here, we have provided proof of principle that shotgun metagenomics can be used to detect and characterise *M. tuberculosis* sequences from sputum samples without culture or target-specific amplification or capture, using an accessible benchtop-sequencing

**Table 3 Species and lineage assignments by phylogenetic placement and spoligotyping.**

| Sample | Phylogenetic placement by pplacer | | Spoligotyping | |
|---|---|---|---|---|
| | Species, lineage, clade | Posterior probability | Lineage | Spoligotype |
| K1 | *M. tuberculosis* Euro-American / Lineage 4 LAM clade | 1 | Euro-American | 1101111111101111111000011111111100001111011 |
| K2 | *M. tuberculosis* Euro-American / Lineage 4 H clade | 1 | Euro-American | 1111111111111111111111111111110100001111111 |
| K3 | *M. africanum* Lineage 6 *M. africanum* clade 2 | 1 | West African 2 | 1111110001111111111000001000011111111101111 |
| K4 | *M. tuberculosis* Euro-American / Lineage 4 H clade | 0.99 | Euro-American | 1111111111111111111111111111110100001111111 |
| K5 | *M. africanum* Lineage 6 *M. africanum* clade 2 | 1 | West African 2 | 1111110001111111111111111111111111111101111 |
| K6 | Not determined | | West African 2 | 1111110001111111111111111111111111111101111 |
| K7 | *M. tuberculosis* Euro-American / Lineage 4 H clade | 0.97 | Euro-American | 1111111111111111111111111111110100001111111 |
| K8 | *M. tuberculosis* Euro-American / Lineage 4 T clade | 1 | Euro-American | 1111110000000000000000000111111100001111111 |

platform, the Illumina MiSeq, and relatively simple DNA extraction, sequencing and bioinformatics protocols.

There are several proven or potential advantages to metagenomics as a diagnostic approach for pulmonary TB. By circumventing the need for culture, it could provide information more quickly than conventional approaches. Even in this proof-of-principle study, for most samples it has provided more detailed information than conventional approaches, including spoligotyping. In addition, it represents an open-ended one-size-fits-all approach that could allow the reunification of TB microbiology with other sputum microbiology, particularly as metagenomics has already been shown to work on other respiratory tract pathogens, including bacteria and viruses (*Lysholm et al., 2012*; *Fischer et al., 2014*). It also aids in the detection of mixed infections (*Chan et al., 2013*; *Koser et al., 2013*), which are clinically important, but hard to recognise (*Shamputa et al., 2004*; *Warren et al., 2004*; *Cohen et al., 2011*; *Wang et al., 2011*; *Hingley-Wilson et al., 2013*).

However, as things stand, there are several important limitations to metagenomics as a diagnostic approach. Our study has been limited to the investigation of smear-positive

**Table 4** IS*6110* profiles.

| Sample | No. reads mapping to IS*6110* | No. reads spanning IS*6110* insertion site | IS*6110* insertion site coordinates | Comments |
|---|---|---|---|---|
| K1 | 11 | 1 | 3480371 | Specific to LAM clade |
| K2 | 199 | 22 | 2610861 (HSI1), 1075947–1075950 (HSI2), 1715974 (HSI3). 212132–212135, 483295–483298, 888787, 1695606, 1986622–1986625, 3120523 | HSI1, HSI2, HSI3 specific to H clade: |
| K3 | 2 | 0 | Not determined | |
| K4 | 6 | 2 | 2610861–2610864 (HSI1) | HSI1 specific to H clade |
| K5 | 4 | 1 | 2631765 | Unique so uninformative |
| K6 | 0 | 0 | Not determined | |
| K7 | 2 | 0 | Not determined | |
| K8 | 5 | 0 | Not determined | |

sputum samples, where a diagnosis can already be obtained quickly and easily by microscopy; considerable improvements in sensitivity are likely to be needed before metagenomics can be made to work on smear-negative culture-positive samples. However, it is worth stressing that smear-positive cases are the most important TB cases in terms of infectivity and severity of disease and rapid, accurate diagnosis and epidemiological investigation of such samples is likely to aid TB control (*Shaw & Wynn-Williams, 1954*; *Colebunders & Bastian, 2000*; *Wang et al., 2008*). Plus, for all our samples, metagenomics goes beyond mere detection of acid-fast bacilli to deliver clinically important information at the level of species and lineage within the *M. tuberculosis* complex.

Surprisingly, metagenomics has not proven quite so informative when applied to contemporary sputum samples as when applied to historical samples, from which we have gained much higher coverage of pathogen genomes, which allowed recognition of phylogenetically informative large sequence polymorphisms (*Chan et al., 2013*; *Kay et al., 2014*). Furthermore, in our hands, sputum metagenomics does not yet deliver sufficient depth of coverage of TB genomes to allow the accurate SNP calling necessary for sequence-based sensitivity testing. It remains unclear whether increased depth of coverage can be achieved by refinements in DNA extraction protocols alone—or whether one might need to sacrifice the speed, simplicity and open-endedness of shotgun metagenomics by incorporating amplification of mycobacterial DNA or cells (i.e., by culture in MGIT tubes (*Koser et al., 2013*)) or by capture of mycobacterial cells or DNA (*Sweeney et al., 2006*; *Bouwman et al., 2012*; *Schuenemann et al., 2013*).

Some have argued that metagenomics is too expensive for routine use (*Köser, Ellington & Peacock, 2014*). However, the same was true of whole-genome sequencing a few years ago; in this study, reagent costs amounted to <£50 per sample. Plus, with minor modifications, we anticipate that DNA extraction could be completed in a few hours of receipt of a sputum sample and sequencing and analysis within a few days. In addition, now that cultured TB isolates are being routinely genome sequenced in many laboratories

(*Koser et al., 2012*; *Kohl et al., 2014*), a catalogue of local TB genomes will be available for comparison with the metagenome-derived genomes, facilitating epidemiological analyses

   With likely future improvements in the ease, throughput and cost-effectiveness of sequencing, twinned with commoditisation of laboratory and informatics workflows, one can foresee a tipping point when a unified automated metagenomics-based workflow might start to compete with the plethora of methods currently in use in the diagnostic microbiology laboratory, while also delivering additional useful information on epidemiology, antimicrobial resistance and pathogen biology.

## ACKNOWLEDGEMENTS

We thank Catherine Okoi for providing an introduction to the MRC Gambia TB laboratory, Abigail Ayorinde for spoligotype analysis and Ousman Secka for sending the DNA extracts to Warwick from The Gambia. We are grateful to the TB field team led by Francis Oko and the Mycobacteriology team at MRC Unit The Gambia. We thank Chrystala Constantinidou, Gemma Kay and Andrew Millard for advice on laboratory and bioinformatics procedures.

### Funding

Support for Emma Doughty's PhD studentship and research costs was provided by Warwick Medical School and MRC Unit, The Gambia. Support for Martin Sergeant's salary was provided by Warwick Medical School. The Enhanced Case Finding project was funded and sponsored by the MRC Unit, The Gambia. The funders had no role in study design, data collection and analysis, decision to publish, or preparation of the manuscript.

### Grant Disclosures

The following grant information was disclosed by the authors:
Warwick Medical School and MRC Unit, The Gambia.

### Competing Interests

The authors declare there are no competing interests.

### Author Contributions

- Emma L. Doughty and Martin J. Sergeant conceived and designed the experiments, performed the experiments, analyzed the data, wrote the paper, prepared figures and/or tables, reviewed drafts of the paper.
- Ifedayo Adetifa conceived and designed the experiments, contributed reagents/materials/analysis tools.
- Martin Antonio conceived and designed the experiments, reviewed drafts of the paper.
- Mark J. Pallen conceived and designed the experiments, wrote the paper, prepared figures and/or tables, reviewed drafts of the paper.

## Human Ethics

The following information was supplied relating to ethical approvals (i.e., approving body and any reference numbers):

The joint Gambia Government/MRC Ethics Committee approved this investigation under reference SCC 1232 and informed written consent was obtained for all participants.

## DNA Deposition

The following information was supplied regarding the deposition of DNA sequences:

Metagenomic sequence reads from this study (excluding those that mapped to the human genome) have been deposited in the European Nucleotide Archive under the following accession numbers: ERS542292, ERS542293, ERS542294, ERS542295, ERS542296, ERS542297, ERS542298, ERS542299.

## Supplemental Information

Supplemental information for this article can be found online at http://dx.doi.org/10.7717/peerj.585#supplemental-information.

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
