# Peer review of "Culture-independent detection and characterisation of Mycobacterium tuberculosis and M. africanum in sputum samples using shotgun metagenomics on a benchtop sequencer"

_PeerJ, doi:10.7717/peerj.585_

## Round 0.1 · original submission · Minor Revisions

Both reviewers are quite positive about the manuscript and in recommend accepting it. One of them has some relatively minor issues that they recommend be addressed and the other points out mostly small text suggestions. I am therefore recommending this as a "minor" revision. Please provide a response to their comments.

Reviewer 1 ·

Basic reporting

No comments

Experimental design

I think more information (in supplementary data) should be provided about the pplacer results in Figure S1- see comments under validity below.

Validity of the findings

This manuscript describes a proof of principle of direct sequencing (by MiSeq) of M. tuberculosis genomes using DNA extracted from smear positive sputa in Africa, comparing results with spoligotyping of cultured strains from the same specimens. Informative metagenomic sequence was generated from seven of the eight specimens tested. SNP phylogenetic placement against reference genomes placed these isolates in the same lineage groups as found by spoligotyping of the corresponding isolates. Bioinformatic analysis of IS6110 insertions allowed supportive IS6110 typing for three specimens.

The conclusion that the authors have shown that shotgun metagenomics can provide diagnostically and phylogenetically useful data in cases of smear positive clinical tuberculosis without DNA amplification or specific capture is justified.

Minor points
I would like to see more information provided on the following:

1. Sample K6, which was the only one not to provide lineage specific data, had the largest percentage of reads aligning to the human genome and the lowest number of bases aligning to H37Rv. Is there any data differentiating this specimen from the others e.g purulent / non purulent/ bloody? This could also indicate an inherent variability in the DNase protocol outcome .

2. The pplacer output to produce Figure S1 should be described in a bit more detail than the legend "Generated from output files from pplacer"

Reviewer 2 ·

Basic reporting

No comments

Experimental design

No comments

Validity of the findings

No comments

Additional comments

This article is overall well written and describes interesting, technically sound data that is important to the fields of Mycobacterium research as well as next generation sequencing for diagnostics. Line numbers would have been helpful. There are some small grammatical errors to be fixed and then the paper is suitable for publication. Specifically:
-in third paragraph of abstract, line 2, unitalicize the parenthesis behind IS6110
-page 5, line 2, "..sequencing for routine use of in some.." delete 'of'
-page 6, paragraph 2, (Kent and Kubica...,1985) is there some text missing or what is the '...' for?
-page 7, paragraph 3, "The stringent protocol which allowed.." delete 'which'
-page 7, paragraph 3, "..into a tab-lineated format.." I would suggest 'tab-delineated'
-page 9, paragraph 1, "..sputum metagenomes that that aligned.." delete one instance of 'that'
-page 9, paragraph 3, "In sample assigned to the LAM clade.." I would suggest "In one sample.." or "In the sample.."
-in legend for Figure 1 and in all the paper titles in reference list, italicize genus and species names
-in the fourth to last reference, something appears to have gone wrong with the formatting and there is extra space

---

## Round 0.2 · accepted · Accept

Thanks for the rapid revisions. Looks like you addressed all the comments.